# Mechanism of Spatiotemporal Air Quality Response to Meteorological Parameters: A National-Scale Analysis in China

**Zhi Qiao** [1],*[ID]**, Feng Wu** [2]**, Xinliang Xu** [3][ID]**, Jin Yang** [4] **and Luo Liu** [5]

[1]    Key Laboratory of Indoor Air Environment Quality Control, School of Environmental Science and Engineering, Tianjin University, Tianjin 300350, China
[2]    Center for Chinese Agricultural Policy, Institute of Geographical Sciences and Natural Resources Research, Chinese Academy of Sciences, Beijing 100101, China
[3]    State Key Laboratory of Resources and Environmental Information Systems, Institute of Geographical Sciences and Natural Resources Research, Chinese Academy of Sciences, Beijing 100101, China
[4]    School of Humanities and Economic Management, China University of Geosciences, Beijing 10083, China
[5]    Guangdong Province Engineering Research Center for Land Information Technology, The College of Natural Resources and Environment, South China Agricultural University, Guangzhou 510642, China
*    Correspondence: qiaozhi@tju.edu.cn; Tel.: +86-22-87402072

**Abstract:** The air quality over China exhibits seasonal and regional variation, resulting from heterogeneity in industrialization, and is highly affected by variability in meteorological conditions. We performed the first national-scale exploration of the relationship between the Air Pollution Index (API) and multiple meteorological parameters in China, using partial correlation and hierarchical cluster analyses. Relative humidity, wind speed, and temperature were the dominant factors influencing air quality year-round, due to their significant effects on pollutant dispersion and/or transformation of pollutants. The response of the API to single or multiple meteorological factors varied among cities and seasons, and a regional clustering of response mechanisms was observed, particularly in winter. Clear north–south differentiation was detected in the mechanisms of API response to relative humidity and wind speed. These findings provide insight into the spatiotemporal variation in air quality sensitivity to meteorological conditions, which will be useful for implementing regional air pollution control strategies.

**Keywords:** Air Pollution Index (API); meteorological parameters; spatiotemperal variation; partial correlation analysis; China

---

## 1. Introduction

Rapid industrialization and urbanization have led to corresponding degrees of air quality degradation [1]. In China, the great economic boom experienced in recent decades has been overshadowed by its notorious atmospheric pollution levels. In 2015, 78.4% of 338 monitored cities exceeded the China National Ambient Air Quality Standards, demonstrating a grim environmental situation despite the implementation of a series of countermeasures [2]. Excessive anthropogenic emissions, dominated by residential energy use, are a major contributor to air pollution in China [3]. However, because these emissions remain almost constant during a given season, air quality is mainly determined by meteorological parameters with multifaceted characteristics that change over various spatiotemporal scales [4]. Therefore, it is essential to explore spatial-temporal relationship between air quality and meteorological conditions if we are to control and reduce pollution with effective measures.

Numerous studies have demonstrated that multiple-scale meteorology plays an important role in air pollution processes [5–9]. Weather conditions, which are described by parameters such as precipitation, relative humidity, and wind speed, have a substantial influence on the formation, dispersion, and deposition of air pollutants [10]. For example, increased absolute humidity and temperature promote gas pollutant partitioning and oxidation reactions, respectively, leading to high concentrations of particulate matter with a diameter of 2.5 μm ($PM_{2.5}$) in Europe [11,12]. Higher wind speeds reduce boundary layer stability and promote the intrusion of $O_3$ to the surface layer in China [13]. Associated with parameters such as temperature and wind speed and direction, synoptic-scale meteorology determines atmospheric stability and circulation at the scale of pollutant accumulation or transport [14]. For example, Chen et al. concluded that synoptic pressure patterns and their evolution were responsible for severe pollution in northern China [15]. Stohl et al. reported a pollution "express highway" involving an explosive cyclone, along which $NO_X$ was transported from North America to Europe [16]. However, previous studies of the correlations between meteorological parameters and air pollution have been restricted to individual air pollutants (e.g., particulate matters, $O_3$, $SO_2$, $NO_X$), and limited investigations have taken the Air Pollution Index (API) into consideration [4]. As an integrated and generalized index, the API offers comprehensive and timely information about air quality. To better describe the relationship between air pollution and multiple-scale meteorology, it is necessary to investigate API responses to changes in meteorological parameters.

Correlations between air pollution and meteorological parameters have been reported in cities worldwide, such as New York [17], Paris [18], Amsterdam [19], Beijing [10,20], Delhi [21], Nagasaki [22], and Seoul [23]. In China, the North China Plain (NCP), Yangtze River Delta (YRD), and Pearl River Delta (PRD) are air pollution research hotspots due to their dense pollution, developed economies, and deteriorated air quality, whereas vast northwestern regions have attracted less attention [24–26]. Spatial variation in these correlations has been reported in revealed studies. Tai et al. reported that nitrate is negatively correlated with temperature in the southeast USA but positively correlated in California and the Great Plains [27]. Otero et al. found that daily maximum temperature is a driver of $O_3$ concentration but has a smaller effect in southern and northern Europe than in central Europe [28]. Zhang et al. reported that days with top 10% $PM_{2.5}$ concentrations coincide with higher relative humidity in Beijing and lower relative humidity in Guangzhou [29]. Because China is the third largest country in the world, representing five climatic zones, variation in meteorological parameters and uneven development likely result in spatial heterogeneity in pollution patterns throughout the country. Previous studies, which have been restricted to single cities or highly polluted regions, have been unable to provide overall insight into regional characteristics on a national scale. Therefore, these non-differentiated national policies that do not consider local weather conditions may not achieve maximum efficiency in controlling atmospheric pollution.

Therefore, we analyzed the spatiotemporal distribution of relationships between the API and multiple meteorological parameters over China and identified the regional clustering of the API response mechanisms in response to meteorological conditions. The results are expected to improve scheduling of pollution forecast system updates and determination of emissions limits in specific cities or regions. Finally, these conclusions are expected to provide a regional strategy for the formulation of national policies.

## 2. Methodology

### 2.1. Data Collection

In this study, 67 cities with valid daily air quality and meteorological monitoring data were used to determine the effects of meteorological parameters on atmospheric pollution. These data cover an 8-year period from January 1, 2005, to December 31, 2012. The locations of these cities are shown in Figure 1.

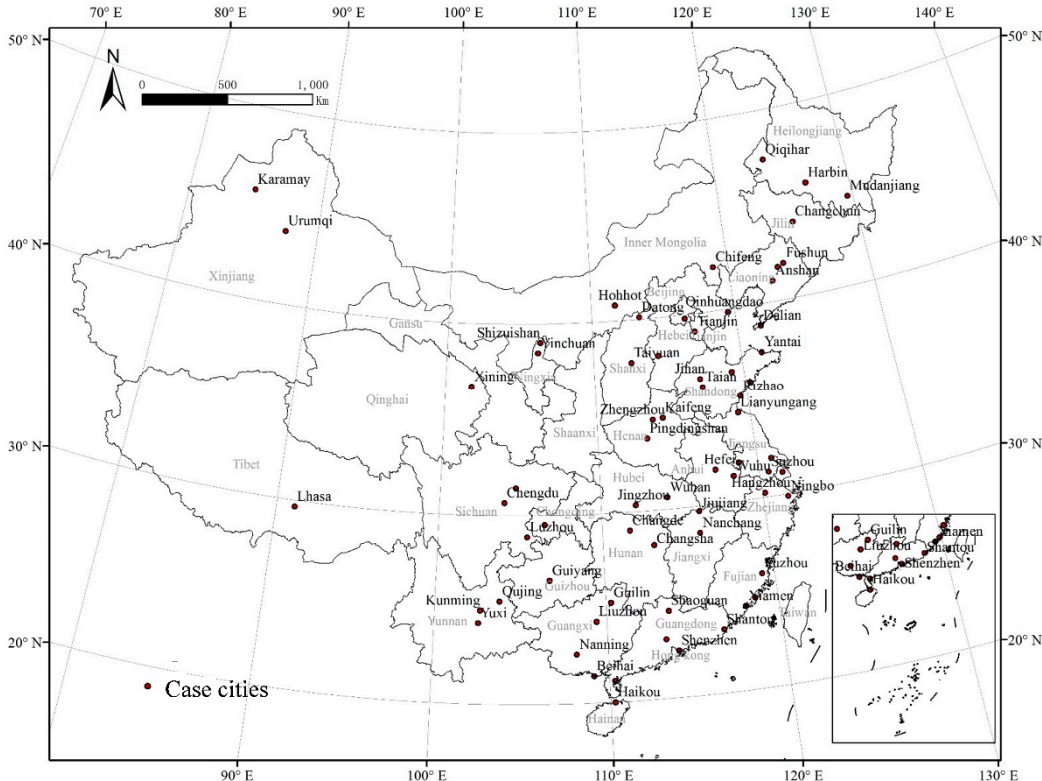

**Figure 1.** Locations of selected cities for analysis.

Air quality was described using the daily API obtained from the Urban Ambient Air Quality Daily Report, which is available on the China Ministry of Environmental Protection website [30]. In China, the API is the official index used to report city's air quality conditions; it is defined as the highest index among those of three critical pollutants: $PM_{10}$, $SO_2$, and $NO_2$ [31]. Sub-API of the three critical pollutants was calculated by a linear interpolation of the reference scale values as given in Table 1 according Equation (1) from the 24 h average mass concentration at each monitoring station.

$$I_p = (PI_{Hi} - PI_{Lo})/(BP_{Hi} - BP_{Lo}) \times \left(C_p - BP_{Lo}\right) + PI_{Lo} \tag{1}$$

where $I_p$ refers to the air pollution sub-index of the three critical pollutants, $p$; $C_p$ is the rounded concentration of the three critical pollutants, $p$; $BP_{Hi}$ and $BP_{Ho}$ is the breakpoint that is greater than $C_p$ and that is less than or equal to $C_p$, respectively; $PI_{Li}$ and $PI_{Lo}$ is API value corresponding to $BP_{Hi}$ and $BP_{Ho}$, respectively.

**Table 1.** Air pollution sub-index levels and their corresponding air pollutant concentrations.

| API Pollution Sub-Index | Air Pollutant Concentrations (μg/m³) | | |
|:---:|:---:|:---:|:---:|
| | $PM_{10}$ 24-h | $SO_2$ 24-h | $NO_2$ 24-h |
| 50 | 50 | 50 | 80 |
| 100 | 150 | 150 | 120 |
| 200 | 350 | 800 | 280 |
| 300 | 420 | 1600 | 565 |
| 400 | 500 | 2100 | 750 |
| 500 | 600 | 2620 | 940 |

The API is defined as the highest index among the three critical pollutants as in Equation (2).

$$API = Max\left(I_{PM_{10}}, \ I_{SO_2}, \ I_{NO_2}\right) \tag{2}$$

Therefore, the daily API dataset contains two parameters: The final API values and the atmospheric pollution type. Larger API values indicate worse air quality, evaluated on a scale of I to V, with V being the worst (Table 2).

**Table 2.** The range of Air Pollution Index (API) and the corresponding air quality levels.

| API | Air Quality Level | Air Quality Description | Health Effects |
|---|---|---|---|
| 0–50 | I | Excellent | Be able to do routine activity normally |
| 51–100 | II | Good | Be able to do routine activity normally |
| 101–150 | III 1 | Slight pollution | A few susceptible may show symptoms |
| 151–200 | III 2 | | Part of healthy population may show symptoms |
| 201–250 | IV 1 | Moderate pollution | Healthy population may show symptoms |
| 251–300 | IV 2 | | The disease symptoms of cardiovascular and respiratory system may aggravate |
| >300 | V | Heavy pollution | Healthy people also will be obviously discomforted |

Weather conditions were represented by surface meteorological data acquired from the National Meteorological Information Center of China [32], including daily precipitation (P) and daily mean values of temperature (T), air pressure (AP), relative humidity (RH), and wind speed (WS).

## 2.2. Partial Correlation Analyses

Partial correlation analyses were used to quantify the relationships between the API and five meteorological parameters (T, AP, RH, P, and WS). According to statistical theory, when a correlation between two variables is influenced by one or more additional variables, the influence of the other variables can be eliminated by introducing partial correlation as a measure of association [33]. Because each meteorological factor may influence ambient air quality to a different degree, partial correlation analyses are appropriate for studying the response of the API to each factor.

For $k + 1$ variables, the partial correlation coefficient (PCC) was calculated as:

$$r_{y1(2)} = \frac{r_{y1} - r_{y2}r_{12}}{\sqrt{\left(1 - r_{y2}^2\right)\left(1 - r_{12}^2\right)}} \tag{3}$$

$$r_{y1(2,3,\cdots,k)} = \frac{r_{y1(2,3,\cdots,k-1)} - r_{yk(2,3,\cdots,k-1)}r_{1k(2,3,\cdots,k-1)}}{\sqrt{\left(1 - r_{yk(2,3,\cdots,k-1)}^2\right)\left(1 - r_{1k(2,3,\cdots,k-1)}^2\right)}} \tag{4}$$

where $r_{y1}$ is the simple correlation coefficient between variable $Y$ (e.g., API) and causative variable $X_1$ (any meteorological parameter). $r_{y2}$ is the simple correlation coefficient between $Y$ and $X_2$ (another meteorological parameter); $r_{12}$ is that between $X_1$ and $X_2$; and $r_{y1(2,3,\cdots,k-1)}$ is the PCC between $Y$ and $X_1$, with the effects of $X_2, X_3, \cdots, X_K$ removed. We estimated the significance of these results based on a two-tailed Student's t-test.

## 2.3. Hierarchical Cluster Analyses

Hierarchical cluster analyses were performed to identify relatively homogeneous groups of cases based on selected characteristics. Through this process, we selected cities with similar correlations between the API and meteorological parameters based on their calculated PCC using Ward's method with Manhattan distance as an analysis measure.

The IBM SPSS Statistics software was used for all analyses described in Sections 3.2 and 3.3. Each of the 12 months was attributed to a single season: Spring (March, April, and May), summer (June, July, and August), autumn (September, October, and November), and winter (December, January, and February); all analyses were performed independently for each of the four seasons in each city.

## 3. Results

### 3.1. Spatiotemporal Characteristics of API

#### 3.1.1. Interannual Variation in API Characteristics

The spatiotemporal characteristics of API varied significantly between years (Table S1). Air quality was generally worse in northern, northeastern, and western China than in other regions. API values were significantly higher in Datong, Taiyuan, Anshan, Beijing, Shijiazhuang, Urumqi, Chifeng, Xining, and Jinan, where steel and coal production were dominant industries. Cities with a vigorous tourism industry in southern China, such as Kunming, Guilin, Beihai, Zhanjiang, and Haikou, generally had better air quality. Prior to 2008, most cities suffered from serious air pollution; however, conditions have gradually improved in recent years, resulting from strict control of anthropogenic emissions, which started during the 2008 Olympic Games in Beijing. National air quality generally improved during 2005–2012, with the annual average API decreasing from 71.93 to 64.18. The greatest improvements were Datong, Taiyuan, Pingdingshan, and Fushun; annual API decreased from 108.03 to 63.47 in Datong during this period.

#### 3.1.2. Seasonal Variation in API Characteristics

In China's major cities, consistent significant seasonal variation in API characteristics was observed (Table S2). Air quality tended to be good in autumn (August, September, and October) and summer (May, June, and July), with average API values of 60.92 and 61.07 during 2005–2012, respectively. Air quality was poorer in winter (November, December, and January) and spring (February, March, and April), with average API values of 79.12 and 72.31, respectively. In spring, pollution was concentrated along the border regions of Shaanxi, Gansu, and Ningxia, as well as in central Inner Mongolia, northwestern Sichuan Province, and Shandong Peninsula. In summer, it was transferred to the Beijing–Tianjin–Hebei urban agglomeration, central Henan urban agglomeration, Yangtze River Delta, and central China, whereas the southeastern coastal areas of the Yunnan–Guizhou Plateau, Tibetan Plateau, and Xinjiang had better air quality. In autumn, spatiotemporal API characteristics were similar to those in summer, except that Inner Mongolia, northeastern China, and southeastern coastal areas that had better air conditions. In winter, air pollution was concentrated in the Beijing–Tianjin–Hebei urban agglomeration, central Henan urban agglomeration, northeastern China, northwestern Sichuan Province, and Xinjiang Uyghur Autonomous Region. Sufficient precipitation in summer is conducive to air purification; monsoons promote pollutant diffusion, which improves air quality. Coal-based central heating north of the Huai River–Qin Mountain line is the main cause of poor air quality in winter.

#### 3.1.3. Variation in Characteristics of Dominant Pollutants

Clean air. The number of clean air days accounted for more than 50% of the total days in the Hainan, Guangdong, Guangxi, and Tibet provinces (Table S3). The maximum number of clean air days reached 80.0% in Zhanjiang, followed by Haikou (79.5%), Beihai (62.3%), Lhasa (59.5%), and Guilin (57.3%). However, the number of clean air days accounted for no more than 18.0% (sometimes as little as 5%) in most northern and northwestern cities, and 20~40% of the total days in southeastern China.

Particulate matter. Particulate matter is one of the most common primary pollutants in most cities in China. Excluding the few cities with clean air, the proportion of days with $PM_{10}$ as the primary pollutant was 60–85% in most cities of China, even reaching 92% in Xining and Jinan.

Chemical pollutants. The proportion of days with chemical pollutants ($SO_2$ or $NO_2$) as the primary pollutant was 0–20% in most cities of China, reaching 27% in Shizuishan, where $SO_2$ was the primary pollutant. $SO_2$ was the primary pollution more frequently in northeastern China, Shanxi, Shandong, Inner Mongolia, and Bohai Bay, due to the types of human activity that predominant in these regions. Cities with $NO_2$ as the primary pollutant are scattered throughout the southeastern coastal areas with the highest proportion in Guangzhou (3.25%), followed by Shenzhen (1.64%).

### 3.2. Relationships between the API and Meteorological Parameters

The PCCs between the API and the five meteorological parameters in different seasons are listed in Supplementary Material Table S4 and shown in Figure 2.

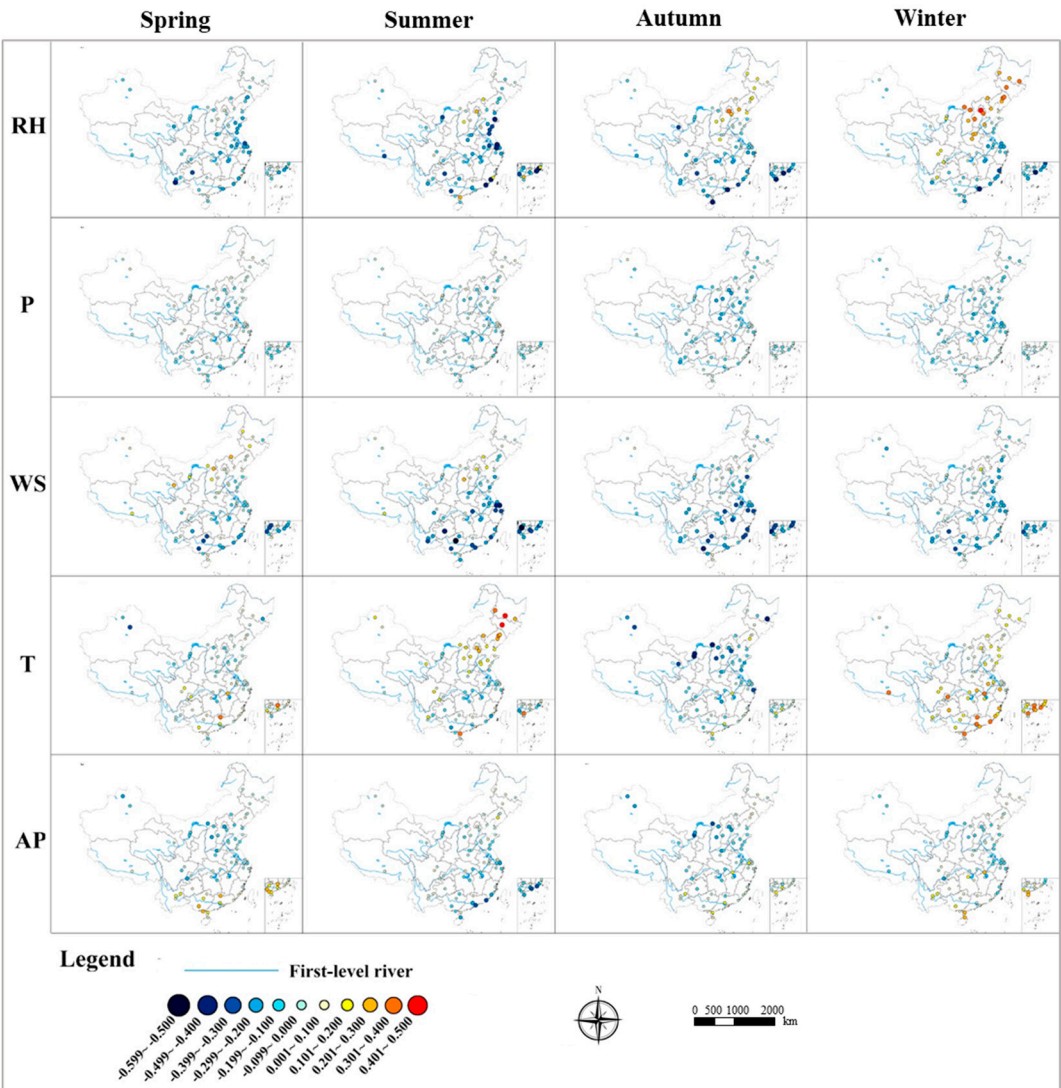

**Figure 2.** Correlations between the Air Pollution Index and five meteorological parameters in each season. Warm and cool colors indicate positive and negative correlations, respectively; larger and smaller symbols indicate larger and smaller PCC magnitudes, respectively.

### 3.2.1. Relationship between the API and Relative Humidity

Among the five meteorological parameters, relative humidity was the most strongly correlated with the API in each season. In spring, there was a negative correlation in almost all cities; the exceptions were five cities with positive but not significant correlations. Thus, we conclude that low API was strongly related to high relative humidity. In summer, there was also a negative correlation in most cities. This correlation was much stronger in coastal Shandong province, the Yangtze River Delta, and southern China. Some cities in northern China showed a positive correlation in spring and a negative correlation in summer. This trend continually intensified from summer to winter, with the region of positive correlation expanding to the area north of the Huai River–Qin Mountain line (also known as the 800 mm isohyet) in winter, when some cities in these regions showed a very strong positive correlation.

Following data transformation, the relationship between relative humidity and the API reflected regional characteristics. API was negatively correlated with relative humidity in southern China, which is the opposite of the trend seen in northern China, particularly in autumn and winter. This result may be related to precipitation, which is generally associated with high humidity. In southern China, more frequent and abundant precipitation due to monsoons increases wet deposition and subsequent scavenging of aerosol particles, dramatically decreasing air pollutant concentrations [34]. This phenomenon explains the reversal in the correlation at 800 mm isohyet. However, in northern China, high relative humidity often occurs following smog episodes, rather than precipitation in autumn and winter, usually accompanied by stagnant atmospheric conditions that inhibit pollutant dispersion and removal. The positive correlation with the API could also be explained by the important influence of relative humidity on photochemical processes and secondary aerosols formation, which is contingent on pollutant type. The concentration of secondary $PM_{10}$ is increased by high relative humidity through the enhancement of heterogeneous reactions of gaseous pollutants (e.g., $NO_2$ and $SO_2$) with primary particles [35]. By contrast, photochemical $O_3$ production decreases under high relative humidity because weak ultraviolet (UV) radiation on humid and cloudy days reduces photochemical reaction productivity [36]. However, $O_3$ is not considered in the calculation of API value; therefore, fewer photochemical reactions can result in higher concentrations of $NO_X$ as precursors of $O_3$, and consequently a higher API [13].

### 3.2.2. Relationship between the API and Precipitation

The API and precipitation were weakly negatively correlated at the national scale and remained generally unchanged year-round. In autumn, this correlation became slightly stronger in northern China; cities in the Yangtze River Basin and coastal region of northern China also showed a stronger correlation in winter.

The negative correlation was weaker than that of relative humidity for two possible reasons. First, unlike the four other parameters examined, which had daily recorded values, there was no precipitation on most days of the year (70.9%) during the study period. Frequent zero value in any variable can cause poor outcomes when calculating PCCs. Second, the dependence of precipitation on relative humidity makes it difficult to separate their effects [36]. As a result, API's correlation with precipitation that should have been manifested might be overshadowed by that with relative humidity to some extent.

### 3.2.3. Relationship between the API and Wind Speed

Wind speed is another dominant meteorological factor significantly affecting air quality throughout the year. In spring, the API and wind speed were positively correlated in northern and northwestern China, and negatively correlated in southeastern China. Nine cities located near the line connecting Changchun to Lhasa showed high, positive PCC values, suggesting that poor air quality in these cities was caused by external pollution sources transported by wind. Conversely, in southeastern China, higher weed speed led to better air quality in most cities. Air pollution sources are mainly local in these cities, with wind enhancing the dilution and diffusion of atmospheric pollutants. In summer, the correlation between the API and wind speed became more strongly negative in southeastern China and more weakly positive in northwestern China. The boundary between a positive and negative correlation shifted to the northwest in summer, and kept moving until autumn, when no significant positive correlation was observed among cities; the overall pattern remained unchanged in winter.

The correlation between wind speed and the API in a given city was highly dependent on whether the main contributor of air pollution was a local or distant source. Wind affects local air quality by altering the dispersion state of the atmosphere [37], but also regional air quality through the transportation of distant air pollutants. The correlation was negative when local sources were dominant, because higher wind speed reduces boundary layer stability, which is conducive to faster dispersion and removal of local air pollutants [13]. By contrast, the correlation was positive when wind

transported pollutants from upstream regions where air quality is poor due to natural (desert or steppe) or anthropogenic sources (intensely industrialized and urbanized areas). The spatiotemporal variation in the correlation observed in most parts of China may explain this phenomenon. In spring, Asian dust storms are carried eastward by prevailing winds and pass over northern China, deteriorating air quality. However, the inflow of comparatively clean maritime air in summer prevailing winds is beneficial to air quality, particularly in coastal areas [34].

### 3.2.4. Relationship between the API and Temperature

Unlike relative humidity and wind speed, the correlation between the API and temperature showed a pattern inversion between seasons. In spring, cities showing a negative correlation were generally located in northern China, whereas those showing a positive correlation were located in southern China. However, the opposite distribution was observed in summer. The API was strongly positively correlated with temperature in northeastern and northern China. However, many cities in the middle and lower Yangtze valley and southern China showed a negative correlation. Similarly, most cities had a negative correlation in autumn and a positive correlation in winter. An apparent gradient from north to south was observed in both autumn and winter PCC maps, indicating that the positive correlation became stronger and the negative correlation became weaker from north to south.

Temperature is the average kinetic energy of molecules in the air; it directly affects the frequency of collision between molecules, on which chemical reactions rely [37], and influences atmospheric conditions through its association with other meteorological parameters such as wind speed [38]. Thus, the correlation between temperature and the API varies according to regional conditions. A positive correlation usually occurs when increasing temperatures result in faster $SO_2$ oxidation to sulfate and carbonaceous aerosol accumulation, which in turn lead to increasing $PM_{10}$ concentrations, particularly in cities where coal combustion is the main source of energy for heating [27]. In agricultural regions, a positive correlation may reflect the temperature dependence of $NH_3$ and $NO_X$ soil emissions, which play key roles in the formation of secondary $PM_{10}$ as precursors of nitrate and ammonium [39]. A negative correlation can be explained by the link between high temperature and vertical and horizontal turbulence, which likely results in greater dispersion and dilution of air pollutants [4]. Given that high temperatures are strongly associated with sunny (extreme UV radiation) days, when photochemical reactions are enhanced, precursors such as $NO_2$ are expected to decrease as they shift to $O_3$ at higher temperatures [13].

### 3.2.5. Relationship between the API and Air Pressure

Generally, the API and air pressure were weakly correlated nationwide throughout the year, especially in winter. The patterns were similar in spring, autumn, and winter. Cities in northern China showed a negative correlation, whereas those in southern China had a positive correlation. Cities in southern China showed a relatively strong positive correlation in spring; cities in southern China and Xinjiang had a relatively stronger negative correlation in spring and autumn. Unlike other seasons, the correlation in summer was moderately negative at the national scale. Only a few cities showed a non-significant positive correlation, and a relatively strong negative correlation was observed in coastal southern China.

Air pressure cannot directly affect chemical reactions in the atmosphere. Instead, it influences the API via its effect on the dispersion state of the atmosphere. A weak dispersion capacity generally indicates an air pollution event. However, both low- and high-pressure systems can promote stable weather conditions, resulting in poor air quality [26]. The migration of a low-pressure system into an area is generally followed by cloudiness or precipitation, whereas high-pressure systems usually lead to sunny, high temperature days; as previously mentioned, the API responds differently to all of these weather conditions. Therefore, in a given city, topographic features and other meteorological factors usually jointly determine whether the correlation between air pressure and the API is positive or negative.

### 3.3. Spatiotemporal Clustering Analyses

Cities were clustered according to significant correlations between multiple meteorological parameters and the API in each season (Figure 3). These clusters indicated differentiation among spatiotemporal distribution characteristics.

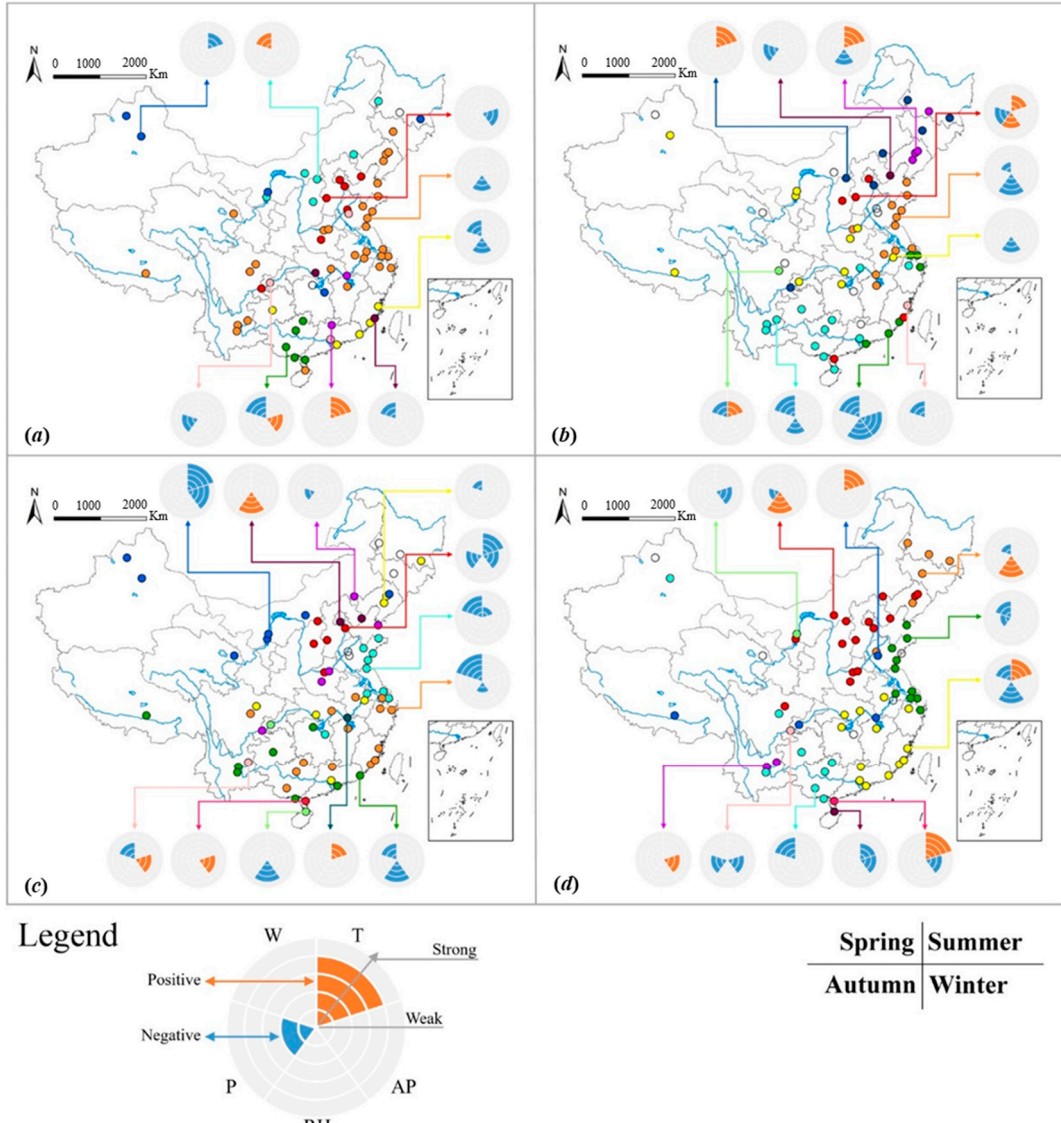

**Figure 3.** Nightingale diagram of cities clustered according to the correlation between the API and dominant meteorological parameters. (**a**) Spring; (**b**) Summer; (**c**) Autumn; (**d**) Winter. Symbols with the same color in each map show the same type of partial correlation. Warm and cool colors indicate positive and negative correlations, respectively. Semi-annulus from center outward represents the degree of correlation, from weakest to strongest.

### 3.3.1. Spatial Clustering Analyses: Spring

In spring, relative humidity was the most strongly correlated with the API, followed by wind speed. A moderate negative correlation was observed in most cities in eastern coastal and southwest inland areas, suggesting that, in spring, higher relative humidity is more likely to result in precipitation, thereby reducing pollutant concentrations by wet deposition over most regions of China.

Unlike relative humidity, which maintained consistent correlation patterns nationwide, the correlation between wind speed and the API showed clear north–south differentiation, demonstrating

the formation mechanism of atmospheric pollution. A moderate to strong negative relationship was found in Fujian and Guangzhou Provinces, and was even stronger in Guangxi Province. One explanation for this result is that the persistent southwest and southeast monsoons, which appear over southern China in spring [40], as well as clean winds from the ocean, may help mitigate air pollution. By contrast, high wind speed led to poor air quality in cities near the border of Inner Mongolia and adjacent provinces to the southeast. The positive correlation in the region occurred only in spring, and may be attributable to dust storms invading this region annually in spring. Such dust storms originate from the Gobi Desert and travel across the Mongolian steppe to northern China; strong winds bring large quantities of particulate pollution to these downwind regions [41], degrading air quality in spring.

Air quality in the Beijing–Tianjin–Hebei urban agglomeration showed a moderate negative correlation with air pressure, which was unusual compared to the rest of China. This phenomenon has been reported in previous studies, such that lower air pressure is associated with serious atmospheric pollution in spring in the region [15]. For example, in Beijing, the probability of air pollution occurrence declines as air pressure increases, and it diminishes only when air pressure is 1025 hPa [24].

This North China Plain phenomenon is associated with its dominant synoptic pollutant conditions in spring. $PM_{10}$ is the major air pollutant in this region [42]; frequent encounters between cold and warm air lead to alternate control by high- and low-pressure systems, resulting in variable weather. When low pressure is the dominant driver, surface temperature increases while wind speed remains relatively low; both of these factors contribute to an increase in $PM_{10}$ concentration.

A correlation between temperature and air quality was detected only in a few scattered cities. Temperature was negatively correlated with the API in four cities in the northern part of China, indicating a link between high temperature and vertical and horizontal turbulence, likely resulting in greater dispersion of air pollutants [4]. Two southern cites, Jiujiang and Shaoguan, showed a positive correlation; both are traditional industrial cities where increasing temperature leads to increased $PM_{10}$.

### 3.3.2. Spatial Clustering Analyses: Summer

In summer, clear north–south spatial variation was observed in selected cities where the API responded to different meteorological parameters. In northeastern and northern China, air quality was positively correlated with temperature, suggesting that summer heat might lead to poor air quality due to increasing concentrations of secondary $PM_{10}$ in these areas, characterized by prosperous agriculture and heavy industry. However, almost every city in southern China had a moderate to strong negative correlation with wind speed and relative humidity; therefore, the low API values in this region were likely associated with strong wind and rainfall brought by the frequent typhoons in summer [43]. Some cities along the eastern coastline also had this correlation, but the effects of wind became weaker from coastal to inland regions. A correlation with wind speed was only detected in cities in the Yangtze River Delta, gradually subsiding and eventually disappearing in most inland cities where API was barely related to relative humidity. These results demonstrate that the summer monsoon significantly affects air quality over southeast China via the transport of clean marine air masses; this influence extends to central China due to northward movement of the monsoon rain belt [40].

Three cities in northern China (Beijing, Shijiazhuang, and Taiyuan) had a positive correlation with relative humidity in summer, a unique phenomenon because most cities retained the negative correlation held in spring. This result suggests that, in these cities, relative humidity was usually not high enough to precipitate moisture in the air in summer; instead, atmospheric moisture was able to enhance the formation of secondary aerosols, leading to higher API values. This explanation is consistent with the negative correlation with precipitation observed in these three cities. Pollutant concentrations then decreased as water vapor condensed into rain.

### 3.3.3. Spatial Clustering Analyses: Autumn

The north–south spatial variation in correlation types evident in summer continued into autumn. The API of most southern cities continued to be governed by relative humidity and wind speed. The negative impact of relative humidity strengthened gradually from north to south, corresponding to the more humid climate. In addition to the clean air mass from the sea, strong southeastward cold-air advection prevailed in autumn, maintaining wind speed as a dominant factor mitigating air pollution in eastern and southern coastal regions [44].

In most northern cities, the correlation with temperature shifted from positive in summer to negative in autumn, accompanied by an emerging negative correlation with air pressure. The increasing role of air pressure can be explained by the increase in activity among both cold and warm air masses in autumn. Large-scale atmospheric circulation played an important role in regional atmospheric structure, increasing the impact of frequent fluctuation in the synoptic pressure system on regional air pollution. The negative correlation between API and temperature and air pressure became stronger in northern inland areas, particularly in northwestern China, where air quality appears to have been greatly affected by the evolution of the pressure system in autumn. This phenomenon was reported in a previous study; a waning cyclone over India and the subsequent onset of the Mongolian anticyclone in autumn have been shown to influence the surface diffusion conditions in northern Xinjiang Uyghur Autonomous Region [45].

Relative humidity and precipitation were also correlated with the API in some northern cities. Compared to summer, some cities in northern China (Beijing and Qinhuangdao) retained a stronger positive correlation with relative humidity, whereas more cities had a negative correlation with precipitation in Shanxi, Hebei, and Henan provinces, indicating that, in autumn, high relative humidity increased the concentration of air pollutants, which were removed as long as humidity was high enough to result in precipitation.

### 3.3.4. Spatial Clustering Analyses: Winter

Air quality is worst in winter due to the combustion of fossil fuels for heating and frequent stagnant weather. In winter, the geographical clustering of correlations between the API and meteorological parameters were clearer than in any other season. Cities were classified into five clusters according to their unique responses to meteorological conditions.

Cities in northern and northeastern China were divided into two clusters both showing a strong to moderate positive correlation with relative humidity. The two clusters differed in that the API was also negatively correlated with precipitation in northern China and wind speed in northeastern China. The mechanism of API response to relative humidity and precipitation in northern China was the same as that observed in other seasons. However, the number of cities showing this type of correlation reached a maximum in winter, consistent with the importance of heating via coal combustion as a contributor to winter air pollution in winter in all cities north of the 800 mm isohyet. High concentrations of $PM_{10}$ and $SO_2$ together with a lack of precipitation facilitated the formation of secondary particles due to increased relative humidity in these dry regions, exacerbating air pollution. The correlation between API and wind speed in northeastern China can be explained by the unfavorable local terrain and snow-covered surfaces in winter, which often result in a stable inversion layer, under which weak winds are conductive to the accumulation of local pollutants [5,6].

Cities in the eastern coastal and southeast inland regions were divided into two clusters. Apart from the moderate negative correlation with wind speed, cities in the Shandong Peninsula and the Yangtze River Delta also showed a weak negative correlation with precipitation, similar to cities in adjacent northern China. However, a positive correlation with temperature and a negative correlation with relative humidity became more significant in cities along the Yangtze River Basin and southern coastline. Wind speed significantly affects pollution levels mainly due to its association with synoptic-scale meteorology, which can determine atmospheric stability and circulation on a larger scale. Previous studies have reported that low wind speed, associated with an L-pattern high-pressure system

in Shanghai, and a relatively uniform pressure field in Nanjing and low-pressure field in Weifang, create disadvantageous dispersion conditions [46]. By contrast, strong winds, usually accompanied by a cold front in winter, are capable of removing pollutants in these regions. Notably, the correlation with relative humidity shifted from positive in the north to negative in the south, suggesting that, in winter, relative humidity in southern cites increased such that precipitation occurred more frequently than in northern cities. Thus, increased relative humidity could enhance wet deposition and aerosol pollutant scavenging. Sea–land breezes and heat-island circulation are also affected by fluctuations in surface temperature and play a role in pollution episodes in southern China.

Another cluster was observed in southwestern inland cities, with the negative correlation between the API and wind speed strengthening along a pathway from the Yunnan–Guizhou Plateau to Guangxi Province. In winter, accompanied by a violent cold front from the north, strong winds effectively scavenged pollutants along this pathway [47]. In areas rich in wind resources, air quality was highly dependent on wind speed in winter, as in spring and summer. Cities in the Sichuan Basin showed different patterns in autumn and winter, perhaps due to their distinct local terrain and emissions intensity.

## 4. Discussion and Conclusions

China has witnessed rapid economic development in the past decades, accompanied by population growth, industrialization and urban expansion [1,48–50]. Such development, however, is compromised by environmental problems, among which atmospheric pollution raises one of the most critical issues [3]. Thus, evaluating air quality in terms of its spatial and temporal dynamics at a relatively large scale is of paramount importance. In this research, monitoring data describing the API and five meteorological parameters for 67 Chinese cities during an eight-year period (2005–2012) were collected and analyzed using several statistical methods, including partial correlation and hierarchical cluster analyses. All analyses focused on the spatiotemporal characteristics of API values and their correlations with meteorological conditions. This study is the first to conduct national-scale analyses of the relationship between meteorological conditions and air quality in China. The API, rather than concentration values of one or several pollutants, was used as an integrated indicator of air quality; the sensitivity of air quality to weather conditions was performed in consideration of multiple meteorological parameters. The results may help to determine and apply strategies for regional air pollution control.

Generally, the API exhibited clear seasonal and regional patterns, influenced by heterogeneity in urbanization and industrialization, as well as variability in meteorological conditions. The correlations were quantified as PCCs, and the results suggest that relative humidity, wind speed, and temperature were the dominant factors influencing air quality throughout the year due to their significant effects on pollutant dispersion and/or transformation. The API was negatively correlated with relative humidity in spring nationwide; the region of positive correlation expanded from northern China in summer to all areas north of the 800 mm isohyet in winter. PCCs between the API and wind speed were positive in spring in northern and northwestern China, but decreased thereafter until all cities showed a negative correlation in autumn. This negative correlation was stronger in southeastern China. The pattern of correlation with temperature was inverted in different seasons. The API was negatively correlated with temperature in northern China and positively in southern China in spring; however, the distribution was completely reversed in summer. Similarly, most cities showed a negative correlation in autumn and a positive correlation in winter. Furthermore, lustering analyses indicated clear north–south differentiation in correlation types, particularly in winter, when selected cities were classified into five clusters according to their unique responses to meteorological conditions.

This study was limited by an uneven distribution of monitoring stations. Sparse data collected in northwestern China may have been insufficient to provide reliable results based on the methods used in this study. The use of daily average values of meteorological parameters eliminated diurnal variation, which could be of substantial importance for air quality. In particular, considerable night–day

temperature discrepancies in some cases could lead to thermal inversion layer events. In addition, we did not explore the effect of wind direction on pollutant concentration. The atmospheric modeling (such as WRF-Chem, CMAx) [51] should be used to simulate the effects of wind direction and wind speed [52] on pollutant dispersion and secondary reactions [53], adding more accurate emissions lists [54], even considering other foreign-based depositions [55] and urban local climate [56–58].

**Supplementary Materials:** The following are available online at http://www.mdpi.com/2071-1050/11/14/3957/s1, Table S1: Interannual variation characteristics of API of 67 cities during 2005–2012, Table S2: Seasonal variation characteristics of API of 67 cities during 2005–2012, Table S3: Variation characteristics of dominant pollutants of 67 cities during 2005–2012; Table S4: PCCs between API and five meteorological parameters in each season.

**Author Contributions:** Conceptualization, Z.Q.; methodology, Z.Q.; software, X.X.; resources, X.X.; data curation, Z.Q. and L.L; visualization, Z.Q.; writing—original draft preparation, Z.Q., F.W., and J.Y.; writing—review and editing, Z.Q.

**Funding:** This research was funded by National Key Research and Development Program of China (Grant #: 2018YFC0213600), Natural Science Foundation of Tianjin City (Grant #: 16JCQNJC08900), National Natural Science Foundation of China (Grants #: 41501472, 41871211, 41571522), and the China Scholarship Council for 1 year's study at Boston University.

**Acknowledgments:** We thank Yixi Li, Xue Rui, Dandan Wang for their assistances with data collection and organization.

**Conflicts of Interest:** The authors declare no conflict of interest.

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
