# Peer review of "Mechanism of Spatiotemporal Air Quality Response to Meteorological Parameters: A National-Scale Analysis in China"

_sustainability, doi:10.3390/su11143957_

Reviewer 1 Report

This paper addresses an important topic: air pollution in China is a serious health concern, and its proper measurement is challenging on many fronts. That said, this is largely an exploratory study to assess whether API and weather correlate appropriately, and the authors do not properly drive these expectations nor state them explicitly. They also do not clarify why – at an exploratory level or otherwise – these issues are important to examine.

I believe that my aforementioned concerns are connected to a crucial issue unaddressed in the paper: the entire study is based on daily API data from China’s Ministry of Environmental Protection with no mention of the problems surrounding these data. While China’s internal API measures have been much improved in recent years, the API data during the sampled period (2005-2012) were likely underestimates of PM10, SO2, and NO2 levels. Given that these data would consistently underestimate China’s air pollution, their use could have been justified by the authors on the basis of data availability or some other reason. (Indeed, it is fortunate that the sampled period does not overlap significantly with the more recent era during which point source estimates have been much improved.) That said, the authors mention nothing about this measurement problem.

Given that many of the observations made in this paper are not rooted sufficiently in the extant literature or in a viable theory, I propose that this study be reframed as a project to assess how API in the pre-reform (i.e. pre-data improvement) era is inconsistent with the reality of meteorological phenomena. Perhaps the authors could addend these data and make an analysis of the structural differences between the pre- and post-reform era to see if there are clear differences between eras in terms of API-weather correlations.

Methodologically, I have tried to decipher the nature of how the cities are listed in the figures, from left to right on the x-axis, but it is still unclear. This may be a function of the authors’ decision to focus on such a large set of Chinese urban areas rather than highlighting the largest (or a set of representative) violators. I do not believe that Figure 2 provides anything meaningful for this study. Figure 3 introduces us to the seasonal variation of China’s air pollution, describing where pollution moves from season to season. Still, each of these cities is being given equal weight in terms of pollution share, and I wonder whether the authors would have been able to communicate a much different story about seasonal air pollution shifts if, for example, the geographic area were reflected in their estimates. Not weighting the scale of air pollution for each of these urban areas effectively assumes that the largest and smallest urban areas are no different; from the perspective of what the authors are attempting to study here – correlations between API and the five meteorological parameters – perhaps that does not matter. It is, however, unexplained clearly. Figure 6 does capture a lot of the necessary information and is useful.

Other issues:

--In the equations, commas are needed between the subscripted, listed numbers; i.e. y1(23…k) should be written as y1(2,3,…,k).

--There is a lack of updated literature. Assessments of Chinese air pollution – relying on atmospheric modeling, point source analysis, or deposition analysis – have become the basis for a cottage industry given not only the problems of air pollution in China but especially because of air pollution flows from Chinese industrial and urban centers to other areas, including foreign-based deposition.

Author Response

Part 1: Response to Reviewer #1

Issue 1: This paper addresses an important topic: air pollution in China is a serious health concern, and its proper measurement is challenging on many fronts. That said, this is largely an exploratory study to assess whether API and weather correlate appropriately, and the authors do not properly drive these expectations nor state them explicitly. They also do not clarify why – at an exploratory level or otherwise – these issues are important to examine.

Response: Accepted and revised.

We thank the reviewers for their comments. Based on the reviewer's comments, we reread our article as a reader and found that we did a lot of quantitative research on the relationships between air quality and meteorological conditions in the original manuscript, and the previous literature research also summarized the research cases of meteorological factors affecting air pollution. However, the significance of the study is not clearly stated.

First, the role of meteorological factors. Air pollution results from a large number of emission sources and high emission intensity, unfavorable weather conditions, terrain. Anthropogenic emissions and meteorological conditions (Xu et al., 2015), which play dominant roles in dispersion, transformation, and removal of air pollutants, interact to create and determine levels of air pollution (Fast et al., 2007). Therefore, it is essential to explore spatial-temporal relationship between air quality and meteorological conditions if we are to control and reduce pollution with effective measures.

Second, research scale. Previous studies, which have been restricted to single cities or highly polluted regions, have been unable to provide overall insight into regional characteristics on a national scale. The neglect of the regional variations makes the relationships between air quality and meteorological factors still unclear in China. The recent years witnessed grim environmental situation despite the implementation of a series of countermeasures in China. Therefore, these non-differentiated national policies that do not consider local weather conditions may not achieve maximum efficiency in controlling atmospheric pollution.

In the revised manuscript, we reorganized these issues and added some key statements. Finally, these conclusions are expected to provide a regional strategy for the formulation of national policies.

Reference

Xu, J., Yan, F., Xie, Y., Wang, F., Wu, J., Fu, Q. Impact of meteorological conditions on a nine-day particulate matter pollution event observed in December 2013, Shanghai, China. Particuology, 2015, 20, 69-79.

Fast, J. D., de Foy, B., Acevedo Rosas, F., Caetano, E., Carmichael, G., Emmons, L., McKenna, D., Mena, M., Skamarock, W., Tie, X., Coulter, R. L., Barnard, J. C., Wiedinmyer, C., and Madronich, S.: A meteorological overview of the MILAGRO field campaigns, Atmos. Chem. Phys., 2007, 7, 2233-2257.

Issue 2: I believe that my aforementioned concerns are connected to a crucial issue unaddressed in the paper: the entire study is based on daily API data from China’s Ministry of Environmental Protection with no mention of the problems surrounding these data. While China’s internal API measures have been much improved in recent years, the API data during the sampled period (2005-2012) were likely underestimates of PM10, SO2, and NO2 levels. Given that these data would consistently underestimate China’s air pollution, their use could have been justified by the authors on the basis of data availability or some other reason. (Indeed, it is fortunate that the sampled period does not overlap significantly with the more recent era during which point source estimates have been much improved.) That said, the authors mention nothing about this measurement problem.

Response: Accepted and revised.

API, which was introduced by US EPA (United States Environmental Protection Agency) for the first time in 1976, was developed and adopted in June 1997 in China to forecast air quality and started to publish by CNEMC (China National Environmental Monitoring Centre) in June 2000. In the revised manuscript, according to the reviewer's comments, we have added the API measurement method in detail as follow.

In China, the API is the official index used to report city’s air quality conditions; it is defined as the highest index among those of three critical pollutants: PM10, SO2, and NO2. Sub-API of the three critical pollutants was calculated by a linear interpolation of the reference scale values as given in Table 1 according the equation 1 from the 24 h average mass concentration at each monitoring station.

,(1)

Where  refers to the air pollution sub-index of the three critical pollutants, ;  is the rounded concentration of the three critical pollutants, ;  and  is the breakpoint that is greater than and that is less than or equal to , respectively;  and  is API value corresponding to  and , respectively.

Table 1. Air pollution sub-index levels and their corresponding air pollutant concentrations.

API pollution   sub-index

Air pollutant concentrations ()

PM10   24-h

SO2   24-h

NO2   24-h

50

50

50

80

100

150

150

120

200

350

800

280

300

420

1600

565

400

500

2100

750

500

600

2620

940

The API is defined as the highest index among the three critical pollutants as the following equation 2.

,

(2)

Therefore, the daily API dataset contains two parameters: the final API values and the atmospheric pollution type. Larger API values indicate worse air quality, evaluated on a scale of I to V, with V being the worst (Table 2).

Table 2. The range of API and the corresponding air quality levels.

API

Air quality level

Air quality   description

Health Effects

0-50

Excellent

Be able to   routine activity normally

51-100

Good

Be able to   routine activity normally

101-150

Ⅲ1

Slight pollution

A few susceptible   may occur symptoms

151-200

Ⅲ2

Part of healthy   population may occur symptoms

201-250

Ⅳ1

Moderate   pollution

Healthy   population may occur symptoms

251-300

Ⅳ2

The disease   symptoms of cardiovascular and respiratory system may aggravate

>300

Heavy pollution

Healthy people   also will be obviously discomfort

In fact, API has been widely used in the study of spatiotemporal pattern of air quality and the influence mechanism of meteorological elements and socioeconomic conditions in specific cities. In addition, the accuracy of the API has also been tested by MODIS AOD (Aerosol Optical Depth) products in Beijing. Perhaps, to a certain extent, as the reviewers said, the API does underestimate the China’s air pollution. But from a national perspective, we still believe the dataset provided by official agencies is credible.

Reference

Jassim, M. S., Coskuner, G. Assessment of spatial variations of particulate matter (PM 10 and PM 2.5) in Bahrain identified by air quality index (AQI). Arabian Journal of Geosciences, 2017, 10(1), 19.

Shen, C., Li, C. An analysis of the intrinsic cross-correlations between API and meteorological elements using DPCCA. Physica A: Statistical Mechanics and its Applications, 2016, 446, 100-109.

Zheng, S., Cao, C. X., Singh, R. P. Comparison of ground based indices (API and AQI) with satellite based aerosol products. Science of the Total Environment, 2014, 488, 398-412.

Issue 3: Given that many of the observations made in this paper are not rooted sufficiently in the extant literature or in a viable theory, I propose that this study be reframed as a project to assess how API in the pre-reform (i.e. pre-data improvement) era is inconsistent with the reality of meteorological phenomena. Perhaps the authors could addend these data and make an analysis of the structural differences between the pre- and post-reform era to see if there are clear differences between eras in terms of API-weather correlations.

Response: Accepted and Clarify.

We must admit that this suggestion is very valuable and would respond to the reviewer from two aspects.

First, data source, i.e. API and AQI (Air Quality Index) representing pre- and post-reform air quality indicators, respectively. Obviously, just as the reviewer knows, API, which was introduced by US EPA (United States Environmental Protection Agency) for the first time in 1976, was developed and adopted in June 1997 in China to forecast air quality and started to publish by CNEMC (China National Environmental Monitoring Centre) in June 2000. It simplifies the concentrations of several air pollutants (SO2, NO2, and PM10) to characterize air pollution level and air quality status in several levels. On February 29, 2012, the MEP (the Ministry of Environmental Protection) of the PRC (the People’s Republic of China) approved the technical regulation on ambient AQI. And the AQI was published instead of API for showing air quality and health implication. The calculation methods of the API and AQI are similar. The API and AQI values could be both calculated by a linear interpolation of the reference scale values and calculated as a maximum among air quality sub-indices of all air pollutants. The air pollutant with the highest concentration is considered as the “responsible pollutant”. However, there are indeed three major differences. (1) In AQI, PM2.5 was included in calculation which is one of the major changes whereas in API calculation PM2.5 was not included. (2) More ground monitoring sites were added to calculate the value of AQI, whereas the value of the API is calculated using only the data of the state-controlled monitoring sites. (3) The time accuracy of AQI values is increased to 1 hour compared to API. Just based on these differences, it seems that AQI has higher precision than API. However, the correlation between AQI and air pollutant was not always found to be reasonably good as compared with API through comparing API and AQI with three different MODIS AOD (Aerosol Optical Depth) products in Beijing (Zheng, Cao, and Singh, 2014).

Second, research objectives. We focus on the spatiotemporal distribution of relationships between the air quality and multiple meteorological parameters at the national scale and identified the regional clustering of the API response mechanisms in response to meteorological conditions. The results are expected to improve scheduling of pollution forecast system updates and determination of emissions limits in specific cities or regions. Therefore, we believe that both API and AQI are only the values of pre- and post-reform air qualities. The most important objective is to explore the relationship between air quality and meteorological factors rather than the air quality itself.

Of course, the reviewer’s suggestion is indeed constructive. If we respectively compare API and AQI with multiple meteorological parameters, the structural differences of API-weather correlations between the pre- and post-reform era could be found. These differences may be caused by the factors such as the adjustment of economic structure and the improvements in environmental management policies, in addition to environmental quality itself. This research is a good inspiration for the future work, and it also involves a huge workload, which requires a combination of factors such as GDP, population, topography, and emission sources. We will also begin this research based on the reviewer's comments.

Reference

Zheng, S., Cao, C. X., Singh, R. P. Comparison of ground based indices (API and AQI) with satellite based aerosol products. Science of the Total Environment, 2014, 488, 398-412.

Issue 4: Methodologically, I have tried to decipher the nature of how the cities are listed in the figures, from left to right on the x-axis, but it is still unclear. This may be a function of the authors’ decision to focus on such a large set of Chinese urban areas rather than highlighting the largest (or a set of representative) violators. I do not believe that Figure 2 provides anything meaningful for this study. Figure 3 introduces us to the seasonal variation of China’s air pollution, describing where pollution moves from season to season. Still, each of these cities is being given equal weight in terms of pollution share, and I wonder whether the authors would have been able to communicate a much different story about seasonal air pollution shifts if, for example, the geographic area were reflected in their estimates. Not weighting the scale of air pollution for each of these urban areas effectively assumes that the largest and smallest urban areas are no different; from the perspective of what the authors are attempting to study here – correlations between API and the five meteorological parameters – perhaps that does not matter. It is, however, unexplained clearly. Figure 6 does capture a lot of the necessary information and is useful.

Response: Accepted and revised.

In the original version, as the reviewers said, some of the figures did look bad. We redraw some figures to enhance their readability. We have redrawn Figure 1 and Figure 5 so that the drawing elements can be seen clearly. For figure 2, 3, and 4, we provided raw data as the table and placed in the supplemental material.

Issue 5: In the equations, commas are needed between the subscripted, listed numbers; i.e. y1(23…k) should be written as y1(2, 3, …, k).

Response: Accepted and revised.

We have modified the statistics based on the reviewer's comment. Please see formula 4 for details.

Issue 6: There is a lack of updated literature. Assessments of Chinese air pollution – relying on atmospheric modeling, point source analysis, or deposition analysis – have become the basis for a cottage industry given not only the problems of air pollution in China but especially because of air pollution flows from Chinese industrial and urban centers to other areas, including foreign-based deposition.

Response: Accepted and revised.

We have added many latest references to confirm the accuracy and significance of our research conclusions.

In addition, we did not explore the effect of wind direction on pollutant concentration. The atmospheric modeling (such as WRF-Chem, CMAx) should be used to simulate the effects of wind direction and wind speed on pollutant dispersion and secondary reactions adding more accurate emissions lists, even considering other foreign-based deposition.

Reference

Feng, T.; Zhao, S.; Bei, N.; Wu, J.; Liu, S.; Li, X.; Liu, L.; Qian, Y.; Yang, Q.C.; Wang, Y.C.; Zhou, W.J.; Gao, J.J.; Li, G.H. Secondary organic aerosol enhanced by increasing atmospheric oxidizing capacity in Beijing–Tianjin–Hebei (BTH), China. Atmos. Chem. Phys. 2019, 19, 7429-7443. https://doi.org/10.5194/acp-19-7429-2019

Chen, S.; Huang, J.; Li, J.; Jia, R.; Jiang, N.; Kang, L.; Ma, X.; Xie, T. Comparison of dust emissions, transport, and deposition between the Taklimakan Desert and Gobi Desert from 2007 to 2011. Sci. China Earth Sci. 2017, 60, 1338. https://doi.org/10.1007/s11430-016-9051-0

Du, J.; Zhang, X.; Huang, T.; Gao, H.; Mo, J.; Mao, X.; Ma, J. Removal of PM2. 5 and secondary inorganic aerosols in the North China Plain by dry deposition. Sci. Total Environ. 2019, 651, 2312-2322. https://doi.org/10.1016/j.scitotenv.2018.10.024

Wang, L.; Fu, J. S.; Wei, W.; Wei, Z.; Meng, C.; Ma, S.; Wang, J. How aerosol direct effects influence the source contributions to PM 2.5 concentrations over Southern Hebei, China in severe winter haze episodes. Front. Env. Sci. Eng. 2018, 12, 13. https://doi.org/10.1007/s11783-018-1014-2

Song, P.; Fei, J.; Li, C.; Huang, X. Simulation of an Asian Dust Storm Event in May 2017. Atmosphere. 2019, 10, 135. https://doi.org/10.3390/atmos10030135

Reviewer 2 Report

This research is quite well conducted using both the appropriate measures and statistical methods. Furthermore, based on the empirical results from a national scale analysis of the associations between air quality and meteorological factors in China, this study draws substantive and meaningful implications and lessons.

Author Response

Thanks for your positive comments.

Reviewer 3 Report

This paper discusses the relation of various meteorological factors on air pollution index (API) at a national level. The main contribution of the article is the analyse of the spatiotemporal distribution of relationships between API and various meteorological parameters over China. Furthermore, the paper also revealed the regional clusters developed based on API’s response mechanisms to considered meteorological conditions. As a consequence, the result of the study reflects on the variation of the spatiotemporal sensitivity of air quality to meteorological conditions, which can be useful for developing more robust regional air pollution control strategies.

The authors are off to a good start; however, this study requires additional consideration of wind direction also as a parameter, mainly to understand the spatiotemporal dynamics with seasons. Alternatively, the authors could include developing a smooth surface map surface of the parameters under consideration over the study area. More information that clarifies and justifies their choice of classical correlation approach rather than spatial correlation methods. The authors could have also used the spatiotemporal kriging approach to reveal a more beneficial outcome of the research.

While the study appears to be sound, the language is unclear, making it difficult to follow. I advise the authors to consider rewriting some portions of the Introduction, methodology and results to make to improve the flow and readability of the text (E.g., Line 85-88). Please try using more natural and not complex sentences. Authors should also consider enhancing the visibility of the maps for the readers. The maps are too small to be read.

Author Response

Issue 1: The authors are off to a good start; however, this study requires additional consideration of wind direction also as a parameter, mainly to understand the spatiotemporal dynamics with seasons. Alternatively, the authors could include developing a smooth surface map surface of the parameters under consideration over the study area. More information that clarifies and justifies their choice of classical correlation approach rather than spatial correlation methods. The authors could have also used the spatiotemporal kriging approach to reveal a more beneficial outcome of the research.

Response: Accepted and clarify.

Thank you very much for reviewer's comments. In the original version of the manuscript, we selected five meteorological factors, i.e. daily precipitation (P), daily mean values of temperature (T), air pressure (AP), relative humidity (RH), and wind speed (WS), in order to explore how they affect air quality both spatially and temporally. Indeed, we have tried to quantitatively study the impact of wind direction on air quality in the initial experiments. However, we failed to find good relationships between wind speed and air quality like several other meteorological parameters on the national scale. Therefore, we did not mention this parameter in the original version of the manuscript. According to the reviewer's suggestions, we have re-examined this issue and reviewed many papers. In these literatures, the authors indeed study the relationship between wind direction and air quality. But these studies are mainly concentrated in a certain city or region (urban agglomeration). We tried again to analyze the relationship between them in many ways, but they do not show good laws or even chaos on the national scale. We thought about one of the most important reasons. If in a particular area, the wind direction is indeed an important meteorological element affecting air quality. This is mainly because the pollution sources of the upwind are fixed there. And the pollution concentrations also depend on the wind speed (because the wind speed can determine the extent of pollutant dispersion). However, for a wide range of studies, if the source of pollution, wind speed and wind direction are simultaneously used as variables, it is difficult to reveal the relationship between wind direction and air quality, especially under the premise of uncertain sources of emissions. Although we have spent a lot of time and energy trying to answer your questions, we regret that we really have no ability to get accurate and quantitative conclusions in this article. We admit that the point is indeed a very good and worth studying. We have collected a list of national emission sources and intend to further study the temporal and spatial relationships between wind direction, wind speed, pollution source and air quality through numerical simulation methods (such as WRF-Chem, CMAx).

Issue 2: While the study appears to be sound, the language is unclear, making it difficult to follow. I advise the authors to consider rewriting some portions of the Introduction, methodology and results to make to improve the flow and readability of the text (E.g., Line 85-88). Please try using more natural and not complex sentences. Authors should also consider enhancing the visibility of the maps for the readers. The maps are too small to be read.

Response: Accepted and revised.

After revising the academic content of the article based on the reviewer's comments, we polished the language. The English in this document has been checked by at least two professional editors, both native speakers of English. For a certificate, please see: http://www.textcheck.com/certificate/4FNSVZ

In addition, we redraw some figures to enhance their readability. We have redrawn Figure 1 and Figure 5 so that the drawing elements can be seen clearly. Figure 2, 3, and 4 have been modified to the original form and placed in the supplemental material.

Round  2

Reviewer 1 Report

I appreciate some of the changes that were made and suspect much of the discussion in the authors' response was left out of the revised paper due to page limits.